# Impact of Air Pollution and COVID-19 Infection on Periprocedural Death in Patients with Acute Coronary Syndrome

**DOI:** 10.3390/ijerph192416654

**Published:** 2022-12-11

**Authors:** Janusz Sielski, Małgorzata Anna Jóźwiak, Karol Kaziród-Wolski, Zbigniew Siudak, Marek Jóźwiak

**Affiliations:** 1Collegium Medicum, Jan Kochanowski University in Kielce, al. IX Wieków Kielc 19A, 25-369 Kielce, Poland; 2European Institute of Post-Graduate Education in Kielce, Duża 21, 25-305 Kielce, Poland; 3Institute of Geography and Environmental Sciences, Jan Kochanowski University in Kielce, Uniwersytecka 7, 25-406 Kielce, Poland

**Keywords:** air pollution, COVID-19 infection, acute coronary syndrome, neural network, left main coronary artery (LMCA) stenosis, critical left anterior descending coronary artery (LAD)

## Abstract

Air pollution and COVID-19 infection affect the pathogenesis of cardiovascular disease. The impact of these factors on the course of ACS treatment is not well defined. The purpose of this study was to evaluate the effects of air pollution, COVID-19 infection, and selected clinical factors on the occurrence of perioperative death in patients with acute coronary syndrome (ACS) by developing a neural network model. This retrospective study included 53,076 patients with ACS from the ORPKI registry (National Registry of Invasive Cardiology Procedures) including 2395 COVID-19 (+) patients and 34,547 COVID-19 (−) patients. The neural network model developed included 57 variables, had high performance in predicting perioperative patient death, and had an error risk of 0.03%. Based on the analysis of the effect of permutation on the variable, the variables with the greatest impact on the prediction of perioperative death were identified to be vascular access, critical stenosis of the left main coronary artery (LMCA) or left anterior descending coronary artery (LAD). Air pollutants and COVID-19 had weaker effects on end-point prediction. The neural network model developed has high performance in predicting the occurrence of perioperative death. Although COVID-19 and air pollutants affect the prediction of perioperative death, the key predictors remain vascular access and critical LMCA or LAD stenosis.

## 1. Introduction

The COVID-19 viral pandemic has been ongoing since 2019, and it is particularly dangerous for older patients with comorbidities and for younger patients with compromised immunity [1].

Acute coronary syndrome (ACS) is one of the most serious medical problems in Poland. During the pandemic, the management of patients with ACS changed in terms of isolating infected patients and adequately protecting uninfected patients in hospitals, as well as protecting medical staff [2,3]. At the beginning of the pandemic, the number of patients hospitalized for ACS including STEMI decreased. The treatment of patients did not change. Such findings were presented in a large study by Campo et al. [4]. Similarly, a significant decrease in the number of procedures performed in ACS was observed in Poland during the pandemic.

There is a high probability of a relationship between the incidence of COVID-19 and air pollution, mainly particulate matter (PM10 or PM2.5) and the heavy metals that it contains [5], because air pollution affects the rate of spread of both chemical (dioxins) and biological (viruses) toxic substances [6]. The health effects of PM10 PM2.5 dust depend on the size of the particles and their chemical composition. PM2.5 can penetrate the deepest parts of the lungs, where it is accumulated or dissolved in biological fluids and then transported throughout the body with the bloodstream [7].

Air pollution in Poland is high. A report by the World Health Organization (WHO) shows that in 2018, out of the 50 European cities analyzed, 36 were in Poland [8].

Air pollution significantly affects parameters of sudden out-of-hospital cardiac arrest [9]. In the analysis of the available literature, numerous papers were encountered evaluating the influence of air pollution factors on patient prognosis after out-of-hospital cardiac arrest [10,11,12]. As demonstrated by Yusuf et al., almost 7 in 10 cases of CAD can be linked to typical risk factors such as high blood pressure, smoking, socioeconomic status, dyslipidemia, poor diet, obesity, diabetes, depression, and air pollution. Thus, air pollution is one of the risk factors for ischemic heart disease [13], as shown by Kuzma et al. in a large clinical study comparing rural and urban areas in Poland on more than 9000 patients with ACS. The risk of air pollution-related ACS was higher in industrial than non-industrial areas. Chronic exposure to air pollution may underlie differences in the short-term effects of particulate air pollution on STEMI incidence. Clearly, the association of air pollution with peri-procedural death is indirect [14]. The risk of peri-procedural death during treatment of ACS was demonstrated by Sielski et al. in a study including 113,456 cases. This risk was significantly higher in patients with history of diabetes, stroke, myocardial infarction, or renal failure. Thus, the current study assumed a possible effect of air pollution on the risk of perioperative death in ACS [15].

However, there is little data on the impact of heavy metal air pollution on the disease course and mortality of COVID-19. Heavy metals contained in PM10 and PM2.5 particulate matter play an important role in the impact of air pollution on human health, especially on the occurrence of cardiovascular diseases [16,17,18,19]. Some authors [9,20,21] in their studies showed statistically significant relationship between the occurrence of ventricular fibrillation (VF) at the arrival of emergency ambulance and the presence of arsenic (As) in the air. A multivariate regression model confirmed the relationship between the occurrence of VF and As and clinical parameters, as well as the relationship between the occurrence of death at the arrival of the ambulance and arsenic exposition.

Periprocedural mortality plays an important role in the analysis of mortality due to ACS [15]. Perioperative death is defined as death occurring from the time of anesthesia until 30 days after the procedure [22]. For this reason, we decided to assess the impact of heavy metal air pollution on periprocedural mortality during ACS in patients infected with COVID-19.

The aim of the study is to determine to what extent air pollution with particulate matter (PM2.5 and PM10) and heavy metals and coexisting COVID-19 infection influence perioperative mortality during ACS in Poland. Another aim is to assess the use of neural networks as a useful method in the analysis of many variables.

## 2. Materials and Methods

### 2.1. Patients Data

The Polish National Registry of Invasive Cardiology Procedures (Ogólnopolski Rejestr Procedur Kardiologii Inwazyjnej, ORPKI) was established in 2004. Since that year, data on the management of ACS have also been collected. Due to the development of IT systems in Poland, the electronic version of the registry has been in operation since January 2014. Currently, there are 161 hemodynamic laboratories in Poland which collect data for the electronic version of the registry, as do invasive cardiology centers in Poland that are reported in the registry [23]. Currently, the management of patients with ACS is based on the guidelines of the Polish Cardiac Society. In general, the most important guidelines include fast transport of a patient with ASC and fast myocardial revascularization [24,25].

Our study group consists of invasively treated patients diagnosed with ACS in Poland and reported to the ORPKI registry during 2020, the year when COVID-19 struck Poland and when both Polish residents and health care workers had their first contact with this pathogen. The study was conducted between 1 January 2020 and 31 December 2020. The group of patients pooled for the analysis—i.e., patients qualified for the treatment ACS and registered in the ORPKI registry—consisted of 53,076 patients. Patients eligible for our analysis were those diagnosed with ACS and qualified for invasive treatment based on the guidelines of the Polish Society of Cardiology. Angiography was performed on all patients qualified for invasive treatment with diagnosed ACS. The study group consisted of patients who were initially diagnosed with acute coronary syndrome and qualified for invasive cardiology procedures. This group included COVID-19 patients as well as those who were not infected. After the COVID-19 test was performed, it became clear who was infected. Then, it was possible to group patients into COVID-19 and non-COVID-19 groups. Both groups were equally exposed to inhaled contaminated air. Only patients registered with complete data for analysis were included; patients with incomplete data were excluded. During the procedure itself, we did not distinguish the sub-periods of the procedure. The time before the invasive procedure was divided into time from onset of coronary pain to first medical contact and time from onset of pain to first balloon inflation or to angiogram. It was not possible to evaluate patients who died several days after the procedure. The ORPKI registry that we used in this manuscript does not collect such data related to death of patients after a hemodynamic procedure. The study group included 34,547 patients who were not infected with COVID-19 (93.8%) and 2395 patients with diagnosed COVID-19 infection (6.2%). Detailed study flowchart is presented on Figure 1.

Patients diagnosed with COVID-19 disease were included in the comparative analysis. Positive results were obtained with an antigen test performed by an emergency medical team, by an appropriate primary care physician, and finally by hospital staff. As this invasive cardiologic procedure must be performed immediately, it was carried out before the final results of the PCR test were received. The study covers the year 2020—the first year of the epidemic. Patients from the pre-vaccination period were analyzed to eliminate the effect of vaccination on the studied parameters. A study of patients with ACS and the impact of vaccination was presented by our research team in another paper [26]. Correlations between acute coronary syndromes and various clinical, peri-procedural, and environmental factors have been studied by our team for several years [15,27,28]. The current work presents for the first time the correlations between environmental factors (air pollution and COVID-19 infection).

Patients who qualified for invasive treatment were asked to sign a declaration of informed consent to the procedure, in accordance with the recommendations of the Helsinki Foundation of 1964. As we used anonymized data from the ORPKI database, the study did not require the approval of the bioethics committee.

### 2.2. Air Pollution Data

Air pollution data obtained from the Main Environmental Protection Inspectorate were analyzed. Air pollutants recorded in 66 air monitoring stations in Poland were analyzed for their content of PM10, PM2.5, and heavy metals such as lead (Pb), nickel (Ni), cadmium (Cd) and arsenic (As) [26]. Daily data for each analyzed pollutant were obtained from the Chief Inspectorate of Environmental Protection for 66 stations in 16 provinces in Poland. The data pertained to the year 2020. The locations of air monitoring stations corresponded in the vast majority to the locations of hemodynamic laboratories to which patients reported. This allowed us to assume that the patients were exposed to the analyzed air pollutants. The amount of air contamination does not have a significant effect on the date the patient reported to the hemodynamic laboratory, as some time must have expired for the contaminants to have an effect. Moreover, the toxic effect of heavy metals depends on many factors, e.g., the rate of metal penetration, distribution in tissues, rate of metal excretion. In addition, cardiovascular diseases with which patients present themselves are the result of a multifactorial response of the body. On the contrary, excess metal and its toxic effects depend on certain factors such as ingestion or inhalation of metal, rate of metal penetration, distribution in tissues and achieved concentration, and finally the rate of metal excretion. Mechanisms of toxicity include inhibition of enzyme activity, protein synthesis, changes in nucleic acid function and changes in cell membrane permeability [29,30].

Knowing the location of the hemodynamics laboratories, sites of air monitoring stations most similar to them were selected, assuming that it was also the place where the patient was exposed to particulate matter and heavy metals. A small proportion of patients were non-local patients who had not been exposed to the effects of air pollution. This is a limitation of the study that was declared in the section “Study Limitations.” Due to the large amount of data and the large number of patients, a neural network constructed for this purpose was used for statistical analysis.

### 2.3. Statistical Analysis

Continuous variables are presented as means and standard deviations (SD) or medians and interquartile ranges (IQR). Categorical variables are presented as numbers and percentages. The normality of the data distribution was checked using Student’s *t*-test. Hertigan’s dip test was used to test for multimodal distributions. Tukey’s test was used to indicate far outliers.

The multiple imputation method was applied to impute the missing data to minimize the effect of missing data on the neural network analysis. The oversampling method was applied to balance the class imbalance in the target variable, which was death during the procedure.

The study sample was randomly divided into two groups: the training group (70%) and the validation group (30%). A feedforward classification that fully connected the multi-layer perceptron neural network with three hidden layers was implemented. All 57 of the variables were added in the input layer. During the learning process, each patient was randomly presented as a new learning case. The algorithm repeatedly tried to match the weights of the variables in order to obtain the best prediction of the outcome. The output layer included one categorical variable, death during the procedure. Three hidden layers were constructed between the input layer and the output layer, which allowed more complex patterns between the input variables and the output variable to be learned. A hyperbolic tangent activation function was used. Binary cross-entropy loss was used for neural network optimization. The neural network was trained with the backpropagation method, using an adaptive stochastic gradient descent algorithm. Our neural network considered 57 variables and was highly effective in predicting patient periprocedural death. The risk of error in this process was 0.03%. The model of the neural network is shown in Figure 2.

The model was evaluated with the area under the receiver operating characteristics curve (AUCROC). After the neural network was trained and evaluated, a permutation feature importance analysis was performed to test which variable had the greatest impact on the neural network model. The results are presented on a graph. The features were ordered from those on which permutation of the feature values had the greatest impact on the neural networks’ binary cross-entropy loss (difference in loss before and after the permutation was largest) to those on which permutation had almost no impact on the neural networks’ loss. In the analysis, a significance level of α ≥ 0.05 was set. The most importance features were those for which the change in binary cross-entropy loss was higher than or equal to 5%.

## 3. Results

Patients hospitalized for ACS in 2020, the first year of the pandemic in Poland, were analyzed. In this group, 2395 patients had a COVID-19 (+) status and 34,547 had a COVID-19 (−) status. The patients in the COVID-19 (+) group were younger and were predominantly male. ACS in the COVID-19 (+) group more often presented clinically as ST-elevation myocardial infarction (STEMI) and much less often as unstable angina (UA) (16,865 (31.8%) vs. 1372 (57.3%), *p* < 0.001). The time from pain onset to first medical contact (180.0 (70.0.540.0) vs. 150.0 (60.0.480.0), *p* < 0.001), time to balloon inflation or angiogram (460.0 (188.0.1348.5) vs. 320.0 (180.0.960.0) *p* < 0.001), and the time from first medical contact to inflation or angiogram (125.0 (60.0.420.0) vs. 120.0 (65.0.240.0) *p* < 0.001) were significantly shorter in the COVID-19 (+) group. We observed 253 (0.68%) peri-procedural deaths in the study group. COVID-19 (+) patients were more likely to be transported directly to the Hemodynamics Laboratory (3350 (11.6) vs. 596 (26.2), *p* < 0.001) and they were more likely to experience sudden cardiac arrest before being admitted to the Hemodynamics Laboratory (441 (1.5) vs. 227 (10.0); *p* < 0.001).

When analyzing the concentrations of basic air pollutants in Poland during the study period, we found that the highest average airborne accumulation of PM10 particulate matter occurred in the Silesian agglomeration, where the average annual concentration of particulate matter was 31.57 µg·m^−3^; the number of ACS cases at that time was 5933 (Figure 3).

The highest average annual concentration of PM2.5 (21.69 µg·m^−3^) was also found in the Silesian agglomeration; the number of ACS cases in this area at that time was 5933 (Figure 4).

The general characteristics of the air pollutants under study are presented in Table 1.

Based on the analysis of the influence of permutation on the variable, the variables with the greatest impact on the prediction of periprocedural death were determined to be vascular access, critical left main coronary artery (LMCA) stenosis, critical left anterior descending coronary artery (LAD) stenosis, unfractionated heparin (UFH), unstable angina, pre-admission cardiac arrest, arterial hypertension, right coronary artery (RCA), age, drug-eluting stent (DES) implantation, kidney disease, COVID-19 infection, diabetes, active smoking, NSTEMI, gender, and number of implanted stents (Figure 5). The quality of the model was confirmed by the area under the ROC curve for the training and validation phases (Appendix A).

## 4. Discussion

The COVID-19 pandemic has caused very significant changes in the functioning of societies around the world as well as in the functioning of global health care. On the other hand, the pandemic does not free us from other diseases or from environmental pollution that can influence the course of these diseases. Due to these special circumstances, we have undertaken a special task: studying the impact of heavy metal air pollution on the incidence of ACS and periprocedural death during invasive treatment procedures in the context of COVID-19 infection. Due to the large amount of medical data from the ORPKI registry and environmental data from the pollution registry, we used a neural network for statistical analysis.

Air pollution is recognized as one of the important factors adversely affecting the severity of COVID-19 infection. It is considered to be one of the risk factors of higher COVID-19 mortality due to its effects on the respiratory system, chronic inflammation, and reduced resistance to infection. Such reports have already been published by Chinese and Italian researchers [5,31,32].

According to Chinese researchers, newly confirmed COVID-19 infections were associated with increased PM2.5, PM10, NO_2_, and O_3_ concentrations in 120 cities in China [33]. There have also been reports in the literature on the influence of heavy metal air pollution on the respiratory system. Cd, mercury (Hg), zinc (Zn), and As play the most prominent roles [34,35,36,37]. There are fewer reports, however, on the influence of these pollutants on the cardiovascular system and ACS [9,38]. In contrast, an analysis of the available literature found no reports on the impact of heavy metal air pollution on periprocedural deaths in patients with and without COVID-19 viral infection.

During the epidemic, there have been many important findings and analyses of the specific behavior of the circulatory system during COVID-19 infection. Shi et al. studied a group of 416 patients infected with COVID-19. They focused on the cardiovascular status of these patients. In the COVID-19 group, elderly patients were found to have a greater number of reported comorbidities—e.g., arterial hypertension—with higher values of high-sensitivity troponins and higher values of serum BNP levels seen in the follow-up [39]. In a large European study, Mafham et al. analyzed a total of 3017 hospitalizations for ACS and COVID-19 in the UK. They reported fewer admissions of patients with ACS and an increase in out-of-hospital deaths during the epidemic [40]. Our study of a total of 53,076 patients hospitalized for ACS in 2020 found that patients with COVID-19 were more likely to present clinically with STEMI and were much less likely to present with UA. Patients with COVID-19 were more likely to be transported directly to the hemodynamic laboratory and were more likely to experience a sudden cardiac arrest before being admitted to the hemodynamic laboratory.

The relationship between the incidence of ACS, periprocedural mortality resulting from this disease, and the incidence of COVID-19 has not been sufficiently reported in the literature [41]. In a relatively small Italian study evaluating the number of hemodynamic procedures in ACS in the province of Terrini in 2020—the first year of the pandemic—they found that the number of hemodynamic procedures decreased. However, the hypothesis that this was related to the reduction of air pollution during lockdown has not been confirmed. Direct associations between the incidence of COVID-19 and heavy metal air pollution and periprocedural mortality in ACS are difficult to find in the literature. However, there are indirect correlations that indicate a significant comorbidity of COVID-19 with obesity, diabetes, hypertension, and cancer. In the case of arsenic, Moon et al. conducted a meta-analysis on its effect on increased cardiovascular morbidity, including ischemic heart disease [42]. Contemporary studies on nickel (Ni) and ACS focus mainly on detecting episodes of acute myocardial ischemia by assessing nickel binding to human albumin [43,44]. However, the influence of airborne PM10 and PM 2.5 on the course of ACS is multifaceted. On the one hand, there are reports of a reduction in the incidence of ACS in metropolitan areas due to the restriction of urban traffic during subsequent pandemic waves [43]. On the other hand, the reduction in the frequency of STEMI and NSTEMI during the COVID-19 epidemic may have resulted from the fear of contacting a physician and having to be transported to the hospital, which is associated with higher mortality in STEMI. For NSTEMI, a correlation was found between lower NO_2_ emissions during lockdown and the number of diagnosed cases of disease [45]. Our analysis of a large group of ACS patients across Poland found that the COVID-19 (+) group, on the day of disease onset, had a higher concentration of PM10 for arsenic, nickel, and lead, as well as a higher concentration of PM10 and PM2.5 overall.

Air pollution significantly affects the cardiovascular system and ACS. In a large Chinese study, Chen R. et al. analyzed a population of 1,292,880 ACS patients from 2239 hospitals in 318 Chinese cities between January 2015 and September 2020. The authors found a significant effect of air pollution especially PM2.5, NO_2_, SO_2_, and CO on the occurrence of ischemic heart disease [46]. These relationships also remain as a function of climate [47]. During the COVID-19 pandemic, interesting correlations were observed in this regard. Reports on the impact of the epidemic on the incidence of ACS are conflicting. Picano et al. report a decrease in ACS in Europe due to the lockdown period [48]. In contrast, results of other study not support the hypothesis of a decrease in ACS due to improved air quality following a lockdown [49].

The analysis of a large number of variables in scientific studies requires the creation of new statistical tools. Logistic regression seems to be a promising tool in this analysis and has been successfully used by authors [28,49]. An even more interesting statistical tool for the analysis of large amounts of data with the use of long-term databases is the creation of neural networks. Niedziela et al. compared the creation and use of a neural network in predicting the risk of death in STEMI patients [50]. Li et al. also used a mathematical model of neural networks to study heavy metal air pollution in major Chinese cities [51]. In our study on the effect of PM2.5, PM10, and heavy metal air pollution on the frequency of COVID-19 infections in patients with ACS, we used a neural network for analysis. The neural network we designed perfectly considers all the numerous variables for analysis and determines with a very high probability the risk of periprocedural death in a patient.

### Study Limitations

A small proportion of patients were non-local patients. In these patients, the place where the air pollution was measured was not the same as their place of residence. It may be significantly similar. At the current stage of obtaining data for the database, it is not possible to determine the proportion of these patients. However, due to its possibility of influencing the results, we include the above fact as a limitation of the study.

Another limitation of the study is the lower number of patients presenting to the hospital with ACS during the COVID-19 pandemic. Thus, the number is different from the time before the epidemic and does not accurately represent the problem of ACS and air pollution. During the pandemic, we also had to deal with different management of infected/suspected patients and their placement in COVID-19 hospitals, which affected ACS treatment time and the course of the disease.

## 5. Conclusions


The clinical factors with the greatest impact on predicting periprocedural death were vascular access, critical LMCA stenosis, and critical LAD stenosis.COVID-19 infection had a strong influence on predicting periprocedural death.Air pollution influences peri-procedural deaths, however, to a lower degree than the other analyzed factors.The neural network that we designed is highly effective in predicting periprocedural death.


## Figures and Tables

**Figure 1 ijerph-19-16654-f001:**
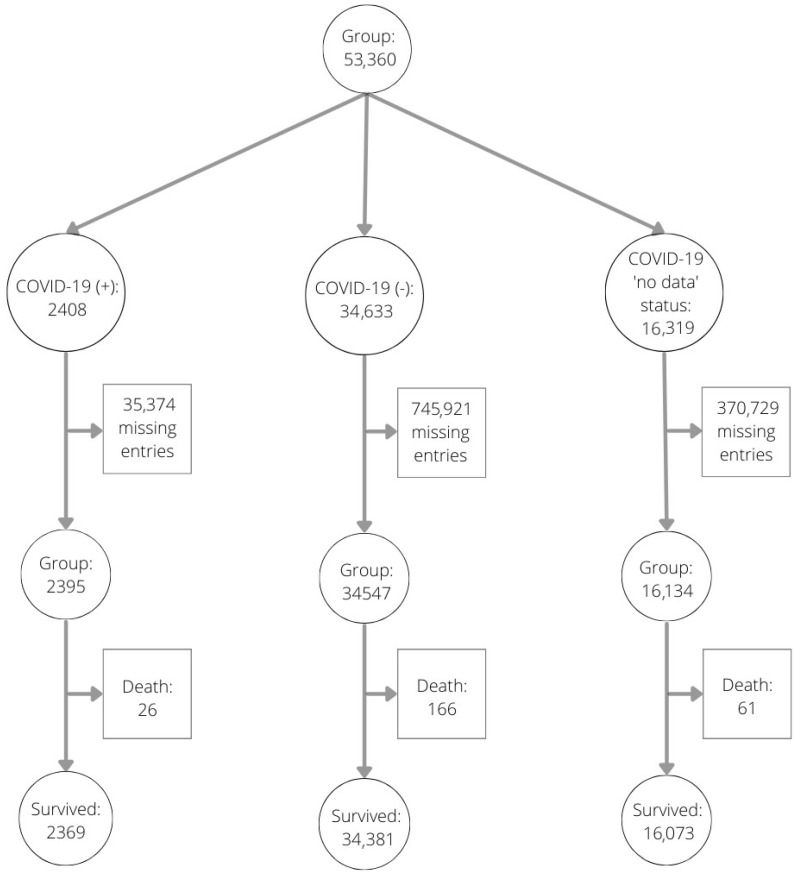
Study flowchart.

**Figure 2 ijerph-19-16654-f002:**
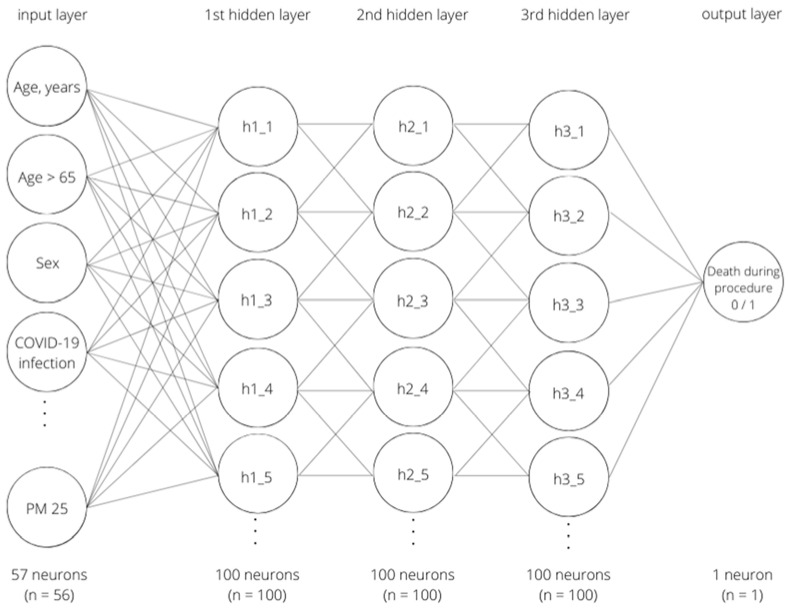
Model of the neural network.

**Figure 3 ijerph-19-16654-f003:**
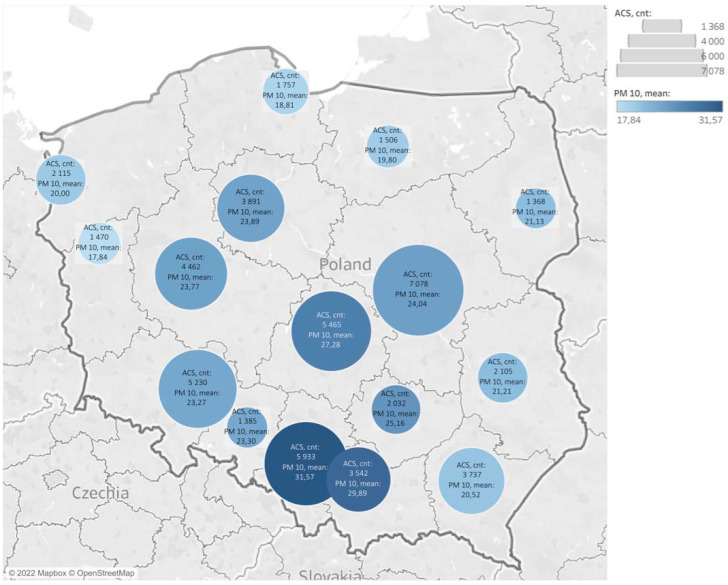
Distribution of acute coronary syndromes and mean PM10 levels in 2020. The circles indicate the accumulation of air pollution in a region. ACS—acute coronary syndrome.

**Figure 4 ijerph-19-16654-f004:**
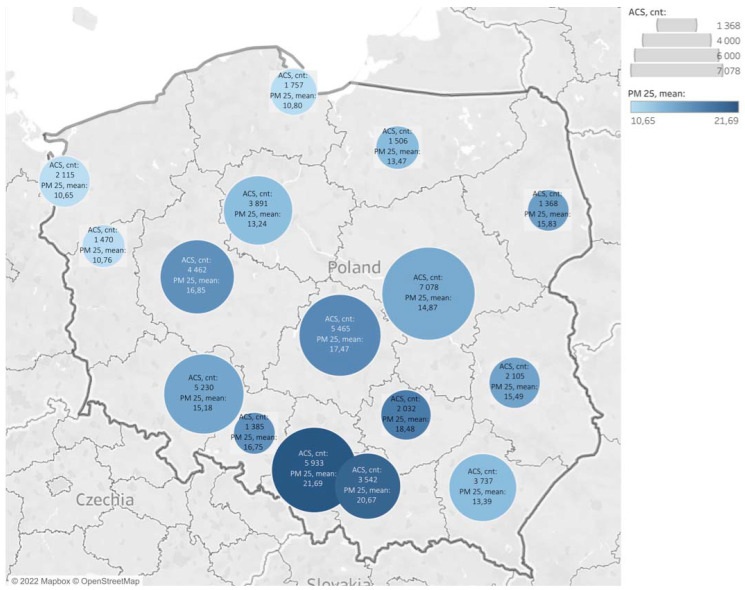
Distribution of acute coronary syndromes and mean PM2.5 levels in 2020. The circles indicate the accumulation of air pollution in a region. ACS—acute coronary syndrome.

**Figure 5 ijerph-19-16654-f005:**
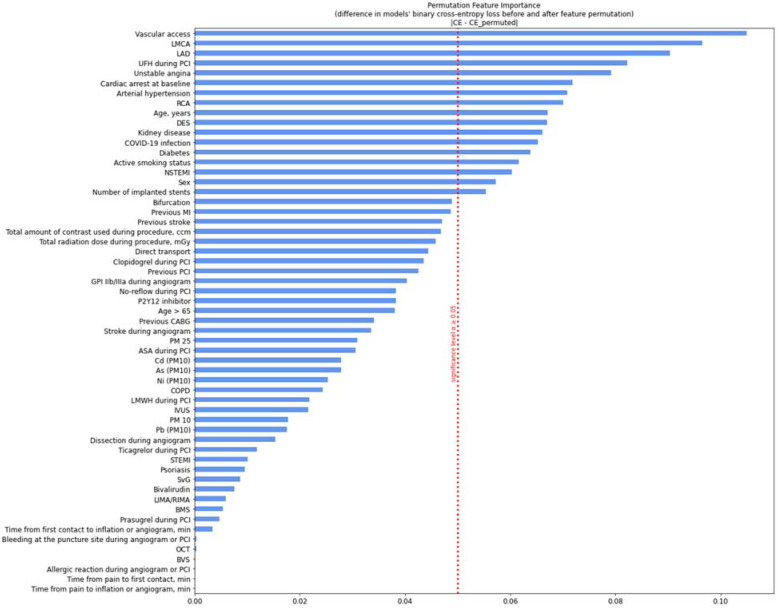
Variables and their impact on the network model based on permutation feature importance analysis. As—arsenic; Cd—Cadmium; Hg—mercury; Zn—zinc; Pb—lead; STEMI—ST-elevation myocardial infarction; NSTEMI—non-ST-elevation myocardial infarction; SD—standard deviation; LMCA—left main coronary artery; LAD—left anterior descending coronary artery; RCA—right coronary artery; DES—drug-eluting stent; PCI—percutaneous coronary intervention; LMWH—low molecular weight heparin; IVUS—intravascular ultrasound; SvG—saphenous vein grafts; BMS—bare metal stent; LIMA—left internal mammary artery; RIMA—right internal mammary artery; OCT—optical coherence tomography; BVS—bioresorbable vascular scaffolds.

**Table 1 ijerph-19-16654-t001:** Descriptive characteristics of air pollution of the analyzed area in 2020.

	As (PM10)	Cd (PM10)	Ni (PM10)	Pb (PM10)	PM10	PM2.5
	ng·m^−3^	ng·m^−3^	µg·m^−3^	µg·m^−3^	µg·m^−3^	ng·m^−3^
records	50,002	50,066	49,808	51,156	52,898	52,607
mean	1.15	0.34	1.76	0.01	24.46	16.05
SD	1.92	0.46	3.16	0.01	14.86	11.25
min	0.10	0.01	0.14	0.0002	2.31	1.00
25%	0.50	0.13	0.50	0.004	14.92	8.90
50%	0.50	0.24	1.12	0.01	20.10	12.68
75%	1.14	0.40	1.97	0.01	29.75	19.53
max	23.20	5.39	91.95	0.27	256.30	166.00

As—arsenic; Cd—Cadmium; Ni—nickel; Pb—lead; SD—standard deviation.

## Data Availability

The data underlying this article will be shared on reasonable request to the corresponding author.

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
