# Peer review of "Impact of Air Pollution and COVID-19 Infection on Periprocedural Death in Patients with Acute Coronary Syndrome"

_ijerph, 2022, doi:10.3390/ijerph192416654_

Round 1

Reviewer 1 Report

Dear Authors,

Thank you for giving me the opportunity to read the article. Although it seems to be well-written in general terms, there are some shortcomings.

1- The introduction is too long, boring the reader, it should be shortened.

2- Patients data; Acceptance and rejection criteria for patient selection are unclear. Which patients were decided to have angiography? What were the inclusion criteria? What were the criteria of those who were not taken? Was angiography performed in each patient?

*Also, reference 29, 3 and 4 in the entry "Patients data" is a descriptive article for a hospital in Poland, not reflecting the essence of the article. It should be removed as it is irrelevant.

* In addition, the start date and end dates of the study are not clearly known.

3-Air pollution data: How did you separate the group with air pollution from the COVID-19 group? COVID_19 patients weren't breathing the same air. How do you overlook the contradiction? Since all patients are breathing the same air, is it inevitable that the COVID-19 group will be significantly higher?

4- Tables should be corrected, not in the place described in the result.

5- There are too many figures, it does not provide the integrity of the meaning.

6- It is not clear what reference 2 represents.Dear Editor,

Author Response

Thank you very much for your insightful review of our work. Please find the responses below. We have marked all changes in red.

  1. The introduction has been shortened and modified.
  2. We have corrected inclusion and exclusion criteria. We have deleted suggested references. Also, we have added study start and end dates.
  3. We have explained how we separate COVID-19 and non-COVID-19 groups.
  4. We have corrected tables.
  5. We have moved Fig 6 and 7 to the supplement
  6. We have deleted this reference.

Reviewer 2 Report

The authors presented the data regarding the impact of air pollution and COVID-19 on periprocedural death among ACS. Several concerns need clarification:

1. Tables 2, the mean  and SD values in different groups are similar, however the p value < 0.05. Is it meaningful clinically? Please elaborate.

2. Based on Figure 4, the air pollution did not influence the periprocedural outcome. However in conlcusion 3 the authors suggest the air pollution to impact periprocedural outcome. Please elaborate.

3. The AUCROC is too good. Please give more explanation! Statistics expert need to be involved.

4. Please clearly described the weakness/limitations of your study.

Author Response

Thank you for this review.

  1. We have removed Table 2 due to low potential for conclusion.
  2. Air pollution does affect peri-procedural deaths, however, less than the other factors shown. Figure 4 is illustrative without conclusion. However, conclusion 3 has been changed.

  3. According to the model's efficiency - for the test set, which consists of 30% of all records from the dataset (31 694 records from 105 646), the neural network model has only 63 false negatives. Below is the confusion matrix for the test set, generated in Python (Scikit-Learn library). array([[15784, 0], [ 63, 15847]], dtype=int64). So this is only 0.199% of misclassified patient's deaths. There is 0,996 True Positive Rate and 0,003 False Positive Rate. Due to this fact, AUCROC curve, taking into account numbers rounding associated with Scikit-Learn "plot_roc_curve" function default behaviour, reflects high model's classification ability.
  4. We have improved limitation section

Reviewer 3 Report

This study analyses and highlights the impact of air pollution and COVID-19 infection on periprocedural death in patients with acute coronary syndrome.

Introduction:

• Give a clear definition of periprocedural death.

Methods:

• This study collects data of the year 2020 only. In this year the COVID-19 disease had a very high mortality rate, it would be appropriate to also evaluate patients with Covid 19 disease in 2021 (after the introduction of the vaccine) to understand if the periprocedural mortality analysed is not affected by the high lethality rate of the virus. • Ensure that correlations are not affected by patient comorbidities and other risk factors that have a significant impact on the proposed outcome.

Discussion:

• The paper states that the air pollution increases the risk of SCA, that the air pollution affects the course of the disease and mortality of COVID-19 and that several studies showed the relationship between the occurrence of death after ventricular fibrillation and arsenic exposition so the association of air pollution and COVID-19 infection with peri-procedural death is indirect. The correlations you want to prove seem not very fluid and not very linear. There are fewer reports, on the influence of the pollutants on the cardiovascular system and ACS. An idea could be to analysethis topic, and then compare it in the pandemic period of the COVID-19 disease.

Author Response

Dear reviewer, thank you for this review. 

Introduction

We have defined periprocedural death. 

Methods

The study covers the year 2020 - the first year of the epidemic. Patients from the pre-vaccination period were analyzed to eliminate the effect of vaccination on the studied parameters. A study of patients with ACS and the impact of vaccination was presented by our research team in another paper (26). Correlations between acute coronary syndromes and various clinical, peri-procedural, and environmental factors have been studied by our team for several years (15, 27, 28). The current work presents for the first time the correlations between environmental factors (air pollution and COVID-19 infection).

Discussion

We have modified discussion as suggested.

Round 2

Reviewer 1 Report

Sevgili Yazarlar,

Görünüşe göre makalede istenen değişiklikler yazarlar tarafından uygulandı. Makaleyi daha elit buldum. Bu olduğu gibi kabul edilebilir.

Saygılarımla.